# Recent Advances in Nattokinase-Enriched Fermented Soybean Foods: A Review

**DOI:** 10.3390/foods11131867

**Published:** 2022-06-24

**Authors:** Danfeng Li, Lizhen Hou, Miao Hu, Yaxin Gao, Zhiliang Tian, Bei Fan, Shuying Li, Fengzhong Wang

**Affiliations:** 1Institute of Food Science and Technology, Chinese Academy of Agricultural Sciences, No. 2 Yuan Ming Yuan West Road, Beijing 100193, China; ldf970920@163.com (D.L.); houlizhen2021@sina.com (L.H.); humiao7890@163.com (M.H.); gaoyx2021@sina.com (Y.G.); tianzhiliang0520@163.com (Z.T.); fanbei@caas.cn (B.F.); 2Key Laboratory of Agro-Products Quality and Safety Control in Storage and Transport Process, Ministry of Agriculture and Rural Affairs, Chinese Academy of Agricultural Sciences, Beijing 100193, China; 3Key Laboratory of Agro-Products Processing, Ministry of Agriculture and Rural Affairs, Chinese Academy of Agricultural Sciences, Beijing 100193, China

**Keywords:** *Bacillus*-fermented food, *Bacillus* spp., fibrinolytic enzyme, nattokinase

## Abstract

With the dramatic increase in mortality of cardiovascular diseases (CVDs) caused by thrombus, this has sparked an interest in seeking more effective thrombolytic drugs or dietary nutriments. The dietary consumption of natto, a traditional *Bacillus*-fermented food (BFF), can reduce the risk of CVDs. Nattokinase (NK), a natural, safe, efficient and cost-effective thrombolytic enzyme, is the most bioactive ingredient in natto. NK has progressively been considered to have potentially beneficial cardiovascular effects. Microbial synthesis is a cost-effective method of producing NK. *Bacillus* spp. are the main production strains. While microbial synthesis of NK has been thoroughly explored, NK yield, activity and stability are the critical restrictions. Multiple optimization strategies are an attempt to tackle the current problems to meet commercial demands. We focus on the recent advances in NK, including fermented soybean foods, production strains, optimization strategies, extraction and purification, activity maintenance, biological functions, and safety assessment of NK. In addition, this review systematically discussed the challenges and prospects of NK in actual application. Due to the continuous exploration and rapid progress of NK, NK is expected to be a natural future alternative to CVDs.

## 1. Introduction

At present, cardiovascular diseases (CVDs) have become the leading cause of human death among various diseases, and are characterized by high morbidity, disability and mortality [1]. Even with the application of the most advanced therapies, most patients are still too late for treatment [2,3]. With the accumulation of experience in the treatment of CVDs and the increase in public awareness of nutrition and health, people gradually realize that dietary is an effective way to prevent and treat CVDs [4].

The consumption of fermented soybean products has shown a significant negative correlation with the risk of CVDs [5,6]. Natto is a traditional soybean food fermented by *Bacillus subtilis* (*Bacillus*-fermented food, BFF) and rich in nattokinase (NK), which is an alkaline serine protease and exhibits strong thrombolytic activity and substrate specificity [7]. NK has a multipotent thrombolytic mechanism (Figure 1) [8,9,10]. For example, NK not only directly degrades fibrin and dissolve thrombi, but also indirectly drives the conversion of endogenous prourokinase to urokinase and plasminogen to plasmin by increasing the level of t-PA (tissue plasminogen activator) and inhibits the level of PAI-1 (plasminogen activator inhibitor) [11,12]. In addition, NK exhibits advantages of high thrombolytic activity, rapid and long duration of action, non-toxic side effects, convenient administration and low production cost, when compared with clinical drugs or other fibrinolytic components, such as urokinase, streptokinase, lumbrokinase, snake venoms and centipede venoms [7,13]. Therefore, NK is expected to be a promising dietary supplement or nutraceutical for the preventive and therapeutic treatment of CVDs [8,14].

This review systematically discusses the recent advances in NK, including fermented soybean foods, production strain sources, optimization strategies, extraction and purification procession, activity maintenance, functional evaluation, and toxicity assessment of NK. Meanwhile, it reveals the serious challenges for NK production by microorganism, as well as the prospects for NK application.

## 2. Fermented Soybean Foods Rich in NK

The consumption of soybean as a food has been recorded for centuries [15]. There are a variety of soybean products, including soybean milk, tofu, bean curd shin, and fermented soybean products. Among the fermented soybean products, such as natto, thua nao, tempeh, knema, douche, gochujang, moromi, chungkookjang, ganjang, doenjang, sieng and pepok are favored by consumers due to their rich flavor, taste and high nutritional value (Table 1) [16,17,18]. Natto is the main representative of leguminous BFFs. There are three types of natto foods, including itohiki-natto, yukiwari-natto and hama-natto, which were extensively consumed in Japan [19]. Fermentation of *Bacillus* spp. results in the extensive degradation of soybean components and the massive production of novel bioactive compounds, such as NK, riboflavin, polyglutamic acid, etc. [15,20]. NK is an important functional factor with thrombolytic activity. In 1906, Sawamura reported the protease activity from natto for the first time [21]. In addition, its fibrinolytic activity was first described by Oshima in 1925 [22]. Miyake et al. succeeded in isolating the protein crystals and determining its amino acid composition [23]. In 1987, Sumi et al. first used the name “NK” in the extraction and isolation of fibrinolytic protein from natto [24]. The protein sequence of NK is highly homologous to the subtilisin NAT (EC 3.4.21.62) produced by *Bacillus*; however, only NK exhibits strong thrombolytic activity and substrate specificity [7].

## 3. Production Strains of NK

Production strains of NK have been isolated from various sources, such as fermented legumes foods, soils, marine water, plants, dairy products, rust, cow dung and others (Table 2), of which BFFs have been proved to be good resources of NK for a long time, such as natto (Figure 2) [39]. *Bacillus* spp. are potential microbial sources of NK, which contains *B. subtilis*, *B. amyloliquefaciens*, *B. licheniformis*, *B. megaterium*, *B. pumilus*, *B.cereus* and others [16,40,41].

*B. subtilis*, as a group of Gram-positive, endospore-forming and rod-shaped bacteria, is generally regarded as a safe (GRAS) microbial producer in fermented legumes foods [16,42]. *B. subtilis* has a short growth cycle, easy fermentation process, high genetic engineering operability, high genetic stability, clear metabolic networks, and no obvious codon preference. Most importantly, it has good protein secretion and expression capacity [42,43]. *B. subtilis* natto, mainly isolated from the fermented vegetable cheese natto, exhibited a multiple-effect fibrinolytic activity of approximately 40 IU/g (wet weight) [24]. *B. subtilis* REVS 12, a production strain of NK, was isolated from a natto food collected in Chennai [44]. Similarly, *B. subtilis* MX-6 from Chinese douche showed the presence of *aprN* encoding NK, which was verified with the polymerase chain reaction (PCR) method [45]. In addition, *B. amyloliquefaciens*, *B. licheniformis*, *B. megaterium*, *B. pumilus*, *B.cereus* and others have also been exploited for NK production [16,40,41]. In addition, other microorganisms are also used as alternatives, such as endophytic *Fusarium* spp., *Zygosaccharomyces rouxii*, *Tetragenococcus halophilus*, and *Pseudomonas* spp. et al. [17,46].

**Table 2 foods-11-01867-t002:** Sources of NK production strains.

Sources			Strains	Ref.
Fermented soybean foods	Natto		*B. subtilis* natto	[24]
			*B. subtilis* natto	[47]
			*B. subtilis* REVS12	[44]
			*B. subtilis* natto B-12	[48]
	Da jang		*B. subtilis* LSSE-62	[49]
	Moromi		*B. subtilis* K2	[17]
	Douchi		*B. subtilis* DC33	[50]
			*B. subtilis* YF38	[51]
			*B. subtilis* MX-6	[45]
			*B. amyloliquefaciens* DC-4	[52]
	Thua nao		*B. subtilis*	[37]
	Chungkukjang		*Bacillus* sp. CK 11-4	[53]
	Doenjang		*B. subtilis* WRL101	[54]
			*Bacillus* sp. DJ-4	[55]
	Gembus		*B. pumilus*	[56]
Soils			*B. subtilis natto* WTC016	[57]
			*B. sublitis* RJAS19	[58]
			*B. subtilis* IMR-NK1	[59]
			*B. subtilis* TKU007	[60]
			*B. subtilis*	[61]
			*Pseudomonas* sp. TKU015	[62]
Marine		Marine water	*B. subtilis*	[63]
		Marine cultures	*B. subtilis* ICTF-1	[64]
Plants		Red alga *Porphyra Dentata*	*B. subtilis* N1	[65]
		The root tissue of *Stemona japonica* (Blume) Miq	*Endophytic Strain* EJS-3	[66]
Dairy products		Bovine milk	*Pseudomonas aeruginosa* CMSS	[67]
		Fermented milk	*B. subtilis* VITMS 2	[68]
Rust			*B. cereus* VITSDVM3	[69]
Cow dung			*Bacillus* spp. IND7	[70]

## 4. Optimization Strategies for NK

Although the microbial production of NK has been intensively studied, yield, activity and stability of NK continue to be the critical restraints on the industrial application of NK. Screening and breeding of NK production strains, culture medium and condition optimization, recombinant expression, and molecular modification are normally used as effective strategies to overcome the above restraints [71,72]. 

### 4.1. Screening and Breeding of NK Production Strains

Strain screening and breeding are proved to be an important way to improve the yield, activity and stability of NK. Production strains of NK are usually screened from traditional BFFs, which were identified to have fibrinolytic capacity [44]. The expression of NK in the wide-type strains is generally low. Therefore, researchers have used various breeding methods to enhance NK expression. Mutation breeding, including physical mutagenesis and chemical mutagenesis, are the most used methods, which can improve the expression yield, activity and stability of NK produced by the production strains. ^60^Co-γ irradiation mutagenesis increased the fibrin hydrolytic activity of the NK production strain by 29.62% and the thermal stability by 82.31% [73]. After the radiation to the protoplasts of *B. subtills* natto by ultraviolet (UV), the viability of NK production strains improved by 16.6%, and the thermal stability increased by 20% after 15 min treatment at 65 °C [74]. NK expression was also enhanced by 2-fold in *P. aeruginosa* CMSS obtained by UV mutagenesis [67]. In addition, the combination of hydroxylamine hydrochloride and UV mutagenesis increased the NK yield of the strain by 68% [75]. In the process of strain screening and breeding, the key to obtain excellent NK production strains is established rapid screening techniques, combined with high throughput screening methods of fibrinolytic activity assays [76].

### 4.2. Culture Medium and Culture Condition Optimization

Culture media composition, culture conditions, and fermentation methods are the main factors influencing NK yield, which are commonly used strategies to improve NK yield.

Optimized medium composition can efficiently facilitate the strains’ growth, which results in the large expression of NK (Table 3). Carbon sources, nitrogen sources, and inorganic salts have a large effect on NK yield of production strains. Carbon sources act as the main substrates to provide energy for the microorganisms to produce NK. In addition, the type and concentration of carbon source can both affect the NK yield. Several studies have shown that glucose was superior to maltose and sucrose as carbon sources for the productivity of NK and increased by 1.23- and 1.25-fold after the replacements, respectively [77]. The NK activity of the medium with yeast extract as the carbon source was 171.1 ± 0.27 U/mL, which was 10 U/mL higher than that of beef extract [68]. The adjustment of the concentration of yeast extract from 3% to 6% in the medium resulted in a 1.12-fold increase in NK activity [78]. Optimal glycerol concentration can also enhance the NK activity. When the concentration of glycerol in the medium was increased by 1%, the NK activity improved by 38.40 U/mL [79]. In addition, rice husk, cassava starch, and other agricultural and sideline products can also be used as carbon sources. When the fermentation medium contained 13.3% rice husk, the NK yield reached 2503.4 IU/g, which was 7-fold of that of the medium without rice husk [80]. In another study, similar NK activity about 1754 U/mL in fermentation media containing tapioca starch and soluble starch was found [81]. In addition, the fibrinolytic activity reached 3682 ± 43 IU/g and 6479 IU/g by using Ginkgo seeds and chestnuts as carbon sources in solid media, respectively [82,83]. 

Nitrogen-containing compounds also can affect NK yield and activity. In addition to the usual nitrogen sources, substrates rich in legume proteins are the main nitrogen sources for NK production, such as soybean milk, soybean protein hydrolysate, soybean peptone, and soybean meal. When soybean milk in the medium increased from 120 to 180 g/L, the NK yield increased by 1.20-fold to 28.3 g/L [77]. Similarly, when the optimized medium consisted of 20 g/L soybean meal, the NK activity increased up to 2174 U/mL, which was 1.24-fold of that of the 10 g/L soybean meal [81]. In addition, the composite nitrogen sources consisting of soybean hydrolysate and soybean meal achieved a higher level of NK activity about 4876 IU/mL, which was 6.5-fold higher than the only medium containing soybean hydrolysate [77]. Notably, some nutrient-rich by-products in food processing are also suitable alternatives of nitrogen sources, such as tofu processing wastewater, cheese whey, and glutamate waste liquid [84,85]. For culture medium with tofu processing wastewater, the NK activity reached 7209.15 ± 195.46 IU/mL and increased by 19.25% compared to soy peptone [84]. Furthermore, the final NK activity was 789.93 U/mL in the low-cost substrate of cheese whey, while the medium cost reduced by 55–60% [85]. 

Inorganic salts can also promote NK yield and enhance its activity [61,68]. Metal ions usually serve as cofactors in NK production. Ca^2+^ can stimulate NK yield; the maximum fibrinolytic activity reached 3787 U/mL at 1.6g/L CaCl_2_ [81]. As the concentration of MgSO_4_ increased from 0 to 0.72%, the NK activity increased by 71% [79]. In addition, when the optimum ratio of Na_2_HPO_4_ and NaH_2_PO_4_ was 4:1, the optimal NK activity was 871.56 IU/mL [86].

It has been demonstrated that suitable culture conditions, such as inoculum size, pH, temperature, and dissolved oxygen, can effectively improve NK activity and yield [87]. Most *Bacillus* species are mesophilic and aerobic strains, tend to thrive well in optimum conditions, and effect NK yield and activity. The optimized NK activity was 3284 ± 58 IU/mL under the culture condition of 30 °C pH 7.0, inoculum amount 2%, and 60 mL of loading volume in a 250 mL conical flask [57]. The NK activity improved by 381 U/mL when adjusting the aeration rate from 1.3 to 10 vvm [81]. Similarly, the NK activity reached 587 U/mL in the fermenter, which was 1.23-fold higher than B shake flask fermentation [78]. In another scale-up experiment in a 100 L fermenter, NK activity reached the maximum of 10,661.97 ± 72.47 IU/mL [84].

The optimization and integration of conditions is the focus of the current research, with higher efficiency, lower production costs, higher NK activity, and more consistent productivity and yields. Various strategies, such as fed-batch and continuous culture, were tested to optimize the NK yield. A supplement with glycerol maintained at a concentration of 20 mL/L resulted in a final NK activity of 7778 17.28 U/mL, 2.6-fold higher than that of the batch [88]. In addition, the highest NK activity reached 654.84 U/mL by fed-batch 3% glycerol and improved by 12% compared to that of the batch culture [78]. 

**Table 3 foods-11-01867-t003:** Culture medium and condition optimization of NK.

Strains	Culture Medium	Culture Methods	Results	Ref.
*B. subtilis* natto WTC016	Luria–Bertani (LB) liquid medium: peptone (10 g/L), yeast extract (5 g/L), NaCl (10 g/L), agar (15 g/L),	Fermented at 30 °C, pH 7.0, and 60 mL of loading volume in 250 mL conical flask for 24 h	3284 ± 58 IU/mL	[57]
*B. subtilis* VITMS2	Sucrose (1%), soybean meal (2%), malt extract (2%), and 10 mM of CaCl_2_, MgSO_4_, Na_2_HPO_4_ and K_2_HPO_4_	Inoculated with 4.0% inoculum, pH 7.0, 30 °C for 48 h	171.1 ± 0.27 U/mL	[68]
*B. subtilis* NDF	Soybean milk (180 g/L) and glucose (105 g/L)	Inoculated with the 5% (*V/V*) of inoculant 3.7 × 10^8^ colony forming unit/mL (CFU/mL) and fed-batch fermented at 30 °C, initial pH 7.0, 600 rpm, 1.0 vvm in 6-L fermenter with 4 L of loading volume for 25 h	10,220 IU/mL	[77]
*B. subtilis* natto	Yeast extract (6%), soy peptone (1.2%), and glycerol (6%)	Inoculated with the 2% volume/volume (*V/V*) spore solution (5.2 ± 0.5 × 10^10^ spores/mL) and fed-batch fermented at 40 °C for 10 h in fermenter	587 U/mL	[78]
*B. subtilis* natto	Soybean flour (16.7%) and rice husk (13.3%) with 70% water content	Solid-state fermentation incubated at 37 °C for 24 h	2503.4 IU/g dry substrate	[80]
*B. subtilis D2*1-8	Cassava starch (20 g/L), soybean meal (10 g/L), K_2_HPO_4_ (3 g/L), KH_2_PO_4_ (1 g/L), MgSO_4_·7H_2_O (3.5 g/L), and CaCl_2_ (0.2 g/L)	2% (*V/V*) inoculated and fermented at 34 °C and 180 rpm for 72 h in 250 mL Erlenmeyer flask with loading 50 mL medium liquid	1754 U/mL	[81]
*B. subtilis* natto 1A752	Ginkgo seeds	Relative humidity 80%, initial water content 73%, at 38 °C, inoculation volume 18% for 38 h	3682 ± 43 IU/g dry substrate	[82]
*B. subtilis* natto	Chestnut	5% (*V/V*) inoculum concentration at 38 °C for 56 h	6479 IU/g dry substrate	[83]
*B. subtilis MTCC* 2616	Tryptone (10 g/L), yeast extract (10 g/L), K_2_HPO_4_·3H_2_O (1 g/L), MgSO_4_·7H_2_O (0.5 g/L), and CaCl_2_·2H_2_O (0.5 g/L).	Fermented at 30 °C and 150 rpm in an orbital shaker for 51 h	789.93 U/mL	[84]
*B. subtilis* GXA-28	The cane molasses contained 35.2% (*w/w*) sucrose, 9.0% (*w/w*) glucose, 15.8% (*w/w*) fructose; the monosodium glutamate waste liquor consisted of 2.0% (*w/w*) glutamate, 0.3% (*w/w*) (NH_4_)_2_SO_4_	2% (*V/V*) inoculum in solid medium at pH 8.0 and 45 °C for 24 h in shallow tray	986 U/g-substrate	[85]
*B. subtilis* natto	Glucose (6.10 g/L), soybean peptone (5.00 g/L), K_2_HPO_4_ (3.00 g/L), MgSO_4_.7H_2_O (0.25 g/L), NaCl (4.27 g/L), CaCl_2_ (0.05 g/L)	Inoculated 5 billion colony-forming units/mL of medium in 37 °C and pH 7.5 for 20 h	55.82 U/mL	[87]
*B. subtilis* 13,932	Glucose (30.868 g/L), tofu processing wastewater (93.669%), MgSO_4_·7H_2_O (1.129 g/L), CaCl_2_ (0.791 g/L)	Fermented at 37 °C, pH 7.0, 70 mL liquid medium, and 200 rpm in 100 L bioreactors	7209.15 ± 195.46 IU/mL	[89]
*B. subtilis* 14,715	25 g of pigeon pea	2% (*V/V*) inoculum (10^7^ CFU/mL) fermented at 35 °C for 32 h	53.03 U/g	[90]
*B. subtilis* natto	LB medium, glycerol (20 mL/L)	Fermented at 37 °C, pH 7.0, in fermenter	7778 ± 17.28 U/mL	[91]
*P. aeruginosa* CMSS	Shrimp shell (1%), KH_2_PO_4_ (0.1%), MgSO_4_ (0.05%)	1% (*V/V*) inoculum at 37 °C, 120 rpm, and pH 7 for 24 h in a shaker	2581 U/ mL	[67]

### 4.3. Recombinant Expression

At present, genetic manipulation is commonly used to improve NK yield. Undoubtedly, recombinant technology offers a promising approach for enhancing NK yield and extensive application. The *aprN* gene encoding NK has been cloned and recombinantly expressed in various suitable expression systems, including microbes, insects, and plants [9,92,93]. Microbes such as *Bacillus* spp., *Escherichia coli* (*E. coli*), lactic acid bacteria (*LAB*), and eucaryotes are always feasible host strains for NK production (Table 4) [9]. *B. subtilis,* as the main source strain of NK, is a food-grade expression and secretory system. Therefore, it became the first choice for NK recombinant expression. However, in addition to NK, it also expresses various natural extracellular proteases, which may substantially degrade the recombinant proteins expressed by the systems [9]. The genetically improved strain of *B. subtilis*, which lacks extracellular proteases, can solve this problem [94]. The NK yield of the recombinant mutant *B. subtilis* LM2531 with multiple deficient lytic genes was significantly increased about 2.6-fold compared with that of prototypical *B. subtilis* 168 [95]. The NK activity and product activity of mutant *B. licheniformis* BL10 with eight deficient proteases increased by 39%, compared to wild-type WX-02 [96]. *E. coli* has a shorter growth cycle, simpler cultural medium, and easier downstream purification processes, which make it a potential way of producing NK. Both the pro-NK and NK were recombinantly expressed in *E. coli*, but the fibrinolytic activity of the recombinant NK was considerably lower than that of wild-type NK [97]. Recombinant *E. coli* DH5α constructed in *B. licheniformis* DW2 as a host for NK resulted in a 6.54-fold increase in *aprE* transcript levels and a 50.53% increase in NK activity [98]. Although NK was expressed largely in *E. coli*, the recombinant protein aggregated to form insoluble and inactive inclusion bodies, which finally reduced NK yield and made it difficult to recover [51,99]. A food-grade *LAB* expression system has enabled the development of oral NK formulations [100]. Recombinant *LAB* can enhance the stability of NK in intestinal fluid, which retained approximately 32% of the fibrinolytic activity after 3 h [100]. Moreover, eukaryotic systems such as *Pichia pastoris*, can also be used for expressing NK [97,101]. However, these hosts exposed the disadvantage of post-translational modification, and long expression times simultaneously. 

In addition, other recombinant expression systems, such as animal and plant, have also been used to express NK. Insects such as *Spodoptera frugiperda* are the normally used animal recombinant expression system. The modified baculovirus expression system successfully expressed NK with 60 U/mL fibrinolytic activity in *S. frugiperda* [92]. Plant systems, named plant molecular farming (PMF), represent hopeful alternative methods for producing recombinant proteins [104,105]. A synthetic gene (*sNK*) was constructed and transiently expressed in melon fruit, with the maximum fibrinolytic activity of 79.30 U/mL [102]. Although PMF-expressed NK had the ability to dissolve fibrin and blood clots in vitro and is not immunogenic by oral intake, PMF application in NK is hampered by some clear restrictions, such as the high cost of downstream extraction processing and low feasibility [106,107].

### 4.4. Molecular Modification

Optimization of the promoter element of *aprN* gene is an effective way to increase NK yields. The expression of *aprN* gene in recombinant B. subtilis can be increased to 643 mg/L by replacing the promoter (from TACAAT to TATAAT) [108]. The exogenous expression of NK by cloning the promoter PnisZ and signal peptide SPUsp in *L. lactis* increased by about 94% [109]. High yields of recombinant NK were also achieved using the tandem promoter PHpaII-PHpaII-PP43 in *B. subtilis* WB800, with a final maximum amplified yield of 816.7 ± 30.0 FU/mL of NK [110]. Besides the optimization of the promoter elements, gene modification is used to improve the fibrinolytic activity of NK. A mutant library of NK has been generated to obtain mutants with higher catalytic efficiency; the S101L, S101F and S101W mutants possessed a 1.45-fold, 1.80-fold and 1.91-fold higher catalytic efficiency, respectively, compared with wild-type NK [111]. In addition, based on several conserved amino acid residues in the determined stereo structure of NK, site-directed mutagenesis was also used to improve the enzymatic properties, such as catalytic activity, stability and antioxidative activity [112]. For a single mutant (I31L) constructed by site-directed mutagenesis, the catalytic efficiency and fibrinolytic activity of this mutant increased about 2- and 1.92-fold relative to the wild-type NK, respectively [113]. The mutant Q59E constructed by site-directed mutagenesis increased specific activity to 10.28 ± 0.5 U/mg and the mutant SY exhibited better stability than other mutants and the wild type under acidic conditions at pH 4, with the activity maintained at 93.3% [114]. Furthermore, 3D model-validated mutants S18D, Q19I, T242Y and Q245W were more stable and less immunogenic than native NK, but without causing any changes in the catalytic site [115]. In another study, for the site-directed mutant M222A optimized amino acid residues Thr220 and Met222 close to the catalytic residue Ser221, the oxidative stability of NK was improved by 10-fold compared to natural NK [112].

## 5. Enrichment Extraction of NK

Enrichment extraction remains an important way to improve NK purity and broaden its application areas. NK is produced by *Bacillus* spp. and secreted extracellularly, which makes it relatively simple to extract. Some traditional protein extraction processes are also applicable to the purification and enrichment of NK, such as protein salting-out, organic solvent distillation, chromatographic chromatography, and dialysis, etc. [9,72]. However, they suffer from operational complexity, inferior efficiency, and low recovery rates [8,39]. Therefore, combined methods are often used in practical production. Microfiltration and ultrafiltration combine to purify NK from fermentation broth with a recovery of 96%, and purity above 92% [72]. Alternatively, NK can be purified by salting-out and then column chromatography [39,116]. NK purified by using a gel filtration (Sephadex G-75) column and hydrophobic interaction chromatography obtained a 56.1-fold activity of the initial activity and the recovery of 43.2% [48]. The purified NK was obtained by ammonium sulfate precipitation and nickel column affinity chromatography with a total NK recovery of 65.2% [91]. The purified NK activity by membrane filtration and Q-Sepharose ion exchange chromatography was 5457 U/mg, with recoveries of 82.45% and 66.91%, respectively [82].

In addition, researchers have developed cheaper, more effective extraction methods and purification techniques, such as reverse micellar extraction, magnetic bead, and three-phase partitioning (TPP) technique [117,118]. The recovery of the AOT (sodium di [2-ethylhexyl] sulfosuccinate)/isooctane reverse micelles system extraction efficiency was 80% with the purification factor of 3.9 [119]. The efficient magnetic poly methyl methacrylate beads immobilized with p-aminobenzamidine have a strong capacity for NK purification with the recovery of 85% and purification factor of 8.7 [120]. NK purified by the TPP technique combined with t-butanol and ammonium sulphate not only showed improved pH and temperature stability, but also achieved 129.5% recovery of NK activity [118]. After graded enrichment, the purified NK was obtained by spray drying or lyophilization, etc.

## 6. Maintenance of NK Activity

The in vivo experiment results confirmed that NK was absorbed through the intestinal tract, and thus took on a functional role [121]. However, free NK is less resistant to the highly acidic pH and was completely inactivated after 1 h in the gastric environment at pH 1.2 [76]. Maintenance of NK activity is the main obstacle to its oral administration effect. Several enzyme immobilization and delivery systems can solve this problem, thus improving the stability and bioavailability of NK.

Enzyme immobilization is commonly used to improve the fibrinolytic activity and stability of NK [122]. The traditional methods are adsorption, embedment, covalent binding, and cross-linking. Immobilized NK can maintain its activity in storage at 4 °C for 25 days using poly hydroxybutyrate physisorption of NK [123]. Embedment usually utilizes biomolecular immobilization. Proteins, polysaccharides, liposomes and biopolymers are all available materials for NK microcapsules. NK activity modified with folic acid-modified chitosan was at least 36% higher than unmodified NK [124]. Phytosterol liposomes (lecithin and phytosterol) encapsulated in NK by a thin film reported an optimal encapsulation rate of 65.25% and NK activity of 2500 U/mL [125]. 

Although the traditional method is simple and inexpensive, there are still certain defects, such as low efficiency and poor stability. Therefore, researchers have developed even more superior enzyme-targeted drug delivery systems [126,127]. A novel chitosan/dialdehyde starch derivative micelle-hydrogel loaded with NK has an NK release rate of 89.4% [128]. Another targeted delivery vehicle using a mesoporous silica/polyglutamic acid peptide dendrimer and NK for electrostatic interaction maximum NK activity of 75% when the ratio was 4/1 [129]. Immobilized NK onto fine magnetic Fe_3_O_4_ nanoparticles with a coupling agent resulted in a higher thrombolytic activity of 91.89%, clearly superior to that of pure NK by 9.03% [130]. In addition to enzyme immobilization, other forms of NK protection and delivery systems are also available matrices, such as tablets and nano-emulsion. NK tablets made from enteric pH-responsive material Eudragit L100-55 and hydroxypropyl cellulose prevented NK inactivation in gastric juice and enabled the controlled release of NK in vitro [131]. High self-double-emulsifying nano emulsions with NK showed a slow-release effect; the encapsulation rate of NK was 86.8 ± 8.2%. The cumulative release after 8h was about 30% [132]. Conjugated poly (lactic-co-glycolic acid) encapsulated NK also works as an effective drug for AD, with a cumulative release rate of 80.23% ± 3.675% over 24 h [133].

## 7. Biological Functions and Pharmaceutical Value of NK

NK can inhibit coagulation, dissolve blood clots, reduce hypertension and hyperlipidemia, promote apoptosis, and regulate autophagy [134,135]. Therefore, NK is highly promising as a natural, safe, effective, and low-cost clinical agent or food supplement for the treatment or prevention of CVDs (Figure 3) [10,15].

### 7.1. Antithrombotic and Fibrinolytic Efficacy of NK

Thrombus is formed by the massive coagulation of fibrin and platelets within blood vessels. The coagulation can be reversed through fibrinolysis. In various vitro and vivo tests, NK has exhibited extremely potent thrombolytic activity with a relatively low risk of hemorrhage, including shortening plasma clot lysis time and reducing the features associated with an increased risk of CVDs [136]. NK delayed thrombosis by over 90% at 75 mg/kg in the model of blood clots, and the safe dose of NK was 15-fold of that of t-PA [137]. Both oral doses and injected NK resulted in a significant reduction by 67.3 to 83.6% in the induced rat models [82]. In a human trial, two NK capsules (2000 FU/capsule) administered orally and daily reduced the features associated with an increased risk of CVDs, plasma fibrinogen levels, factor VII and factor VIII by 9%, 14% and 17%, respectively, for patients with CVDs [12]. In another study, factor VIII activity decreased by an average of 7.4% and 7.6% after 4 and 6 hours of oral 2000 FU NK, respectively, compared to the initial time [138].

### 7.2. Anti-Atherosclerotic Effect of NK

The underlying pathological change common to various CVDs is atherosclerosis (AS), which results in heart disease and stroke. AS is a group of lesions based on disorders of lipid metabolism with some symptoms such as thickening and hardening of the arterial blood vessel walls. NK suppresses intimal thickening through its synergistic antioxidant and anti-apoptotic effects. In addition, NK prevents AS by reducing lipid peroxidation, improving lipid metabolism, and inhibiting low-density lipo-protein (LDL) oxidation [139]. NK treatment resulted in a significant reduction in intima-media thickness and carotid plaque size by 36% and 10.62%, respectively, for patients. Therefore, NK is promised to be a viable alternative therapy for atherosclerotic plaque-induced CVDs and stroke [140].

### 7.3. Hyperlipidemia Reducing Effects of NK

Hyperlipidemia (HLP) is a condition in which blood lipid levels are too high, directly causing several serious diseases such as atherosclerosis. NK prevented HLP by reducing lipid peroxidation and improving lipid metabolism. In a clinical study, total serum cholesterol (TC) levels decreased by 6.8% after 8 weeks in patients with primary hypercholesterolemia treated with 4000FU NK, compared to the placebo group [141]. In addition, the combination of NK with red yeast rice extract had a hypolipidemic effect for patients with hyperlipidemia, which decreased triglycerides (TG), TC and low-density lipoprotein cholesterol (LDL-C) by 15%, 25%, 41%, respectively, and increased HDL-C by 7.5% [142]. In male white rabbit models, the combination of NK and red ginseng reduced the serum cholesteryl ester transfer protein activity by an average of 22%, compared with the high cholesterol groups [143].

### 7.4. Antihypertensive Effects of NK

Hypertension is characterized by an increase in arterial blood pressure (BP) in the body circulation, which is the most common chronic disease and the most important risk factor for CVDs. NK supplementation reduces systolic and diastolic blood pressure through the cleavage of fibrinogen in plasma, suggesting its role in the prevention and treatment of hypertension [144]. In addition, NK was considered a relatively strong angiotensin-converting enzyme inhibitor (ACEI), which may be attributed to the reduction in BP [14]. In a vitro coagulation lysis assay, the ACEI activity of NK reached a maximum of 87.45% at the NK concentration of 4.8 mg/mL [68]. In addition, an 8-week intake of NK decreased the average baseline diastolic BP from 86 to 81 ± 2.5 mmHg in North American hypertensives [145].

### 7.5. Neuroprotective Effects of NK

Degenerative diseases of the central nervous system are a group of diseases caused by chronic degenerative tissue degeneration. The main pathological feature is the extracellular deposition of β-amyloid. Memory dysfunction, cognitive function deficit, language function decline, etc., are common clinical symptoms. In vivo and vitro studies demonstrated that the neuroprotective effect of NK was associated with inhibiting β-amyloid deposition, promoting proteolysis, anti-inflammatory and antiapoptotic effects [146]. NK significantly reduced the cerebral infarct volume by approximately 61% in photothrombotic stroke patients [147]. NK also played an important role in the treatment of amyloid-related diseases, such as Alzheimer’s disease (AD) [148]. In rat models, NK can exhibit a positive effect on regulating specific factors in the AD pathway, including enhancing impaired learning, memory capability, and effective suppressing of β-amyloid.

### 7.6. Other Diseases

NK is also an effective treatment for other diseases, due to its protease properties. NK can be an alternative treatment strategy for retinal neovascularization by the anti-angiogenic effect [149]. NK can treat patients with proliferative vitreoretinal disease by hydrolyzing collagen fiber [150]. NK also treats chronic rhinosinusitis and asthma by hydrolyzing fibrin in nasal polyps of patients with chronic sinusitis [151]. In addition, NK improved survival and inhibited tumor growth in mice with liver cancer [152]. 

## 8. Toxicity Assessment of NK

Given the growing public interest in the prospective health benefits of NK, it is important to assess the potential toxicity and overall safety of NK. NK as a bioactive ingredient of fermented food has been consumed for over a thousand years. NK is virtually safe when administered at the commended dose (200 mg per day) and does not show any signs of toxicity in the histopathological examination of organs and tissues. In a 14-day acute toxicity study, no significant signs of adverse effects or mortality were observed in rats gavaged with 2000 mg/kg of NK and those that continued gavaging up to 90 days also showed no abnormalities [153]. Similarly, there was no mortality, adverse clinical signs, any tissue damage, or gross pathological findings in rat models exposed to single gavage doses of NK at 1000 and 2000 mg/kg during the acute 28-day and 90-day repeated dose toxicity study period. Moreover, in human trials, healthy volunteers taking daily oral NK (10 mg/kg) for 28 days also did not exhibit any adverse effects [154]. In animal and human trials, the same physiological effects indicate the bioequivalence of NK. The safety of NK still needs to be thought about and studied, including the effects of doses, administration time, toxicity and genotoxicity [144,154].

## 9. Conclusions and Prospects

Nattokinase (NK) is a natural, safe, efficient and cost-effective thrombolytic enzyme, which exhibits potentially beneficial cardiovascular effects. This review focused on NK-enriched fermented soybean foods, and more systematically, elaborated on microbial synthesis, extraction and purification, activity maintenance, thrombolytic activity, and safety assessment of NK. Undoubtedly, microbial synthesis by *Bacillus* spp. is a commonly used method of producing NK. The yield, fibrinolytic activity and bioavailability of NK directly determined its industrial applications. At present, with the gradual increase in practical demand, the public are becoming more interested in the health-related potential of NK and fermented soybean foods enriched with NK. Therefore, although microbial synthesis is an extremely cost-effective way to produce NK, its yield, activity and stability are the primary constraints of industrial applications. The improvement of NK yield, fibrinolytic activity and stability, including potential high throughput microbial producer screening, optimization of fermentation medium and conditions, recombinant expression, and molecular modification, can increase the expression of NK. Moreover, the novel enrichment and purification techniques could enhance the activity and stability of NK. In addition, the development of various stable deliveries and formulations of NK could also meet commercial demands. The biological value and functional activity of NK still need to be validated to enhance the bioavailability. Further studies will certainly focuse on multiple theoretical studies and advanced technologies of NK microbial synthesis and contributions to CVDs.

## Figures and Tables

**Figure 1 foods-11-01867-f001:**
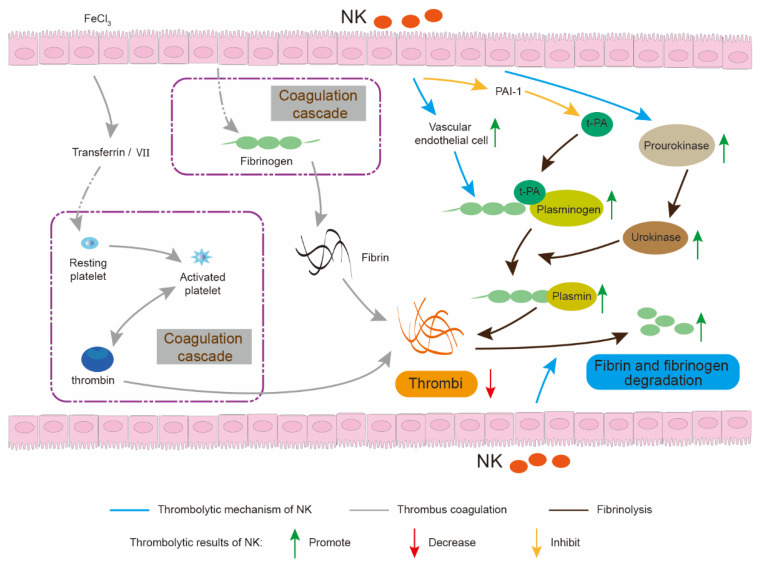
Thrombolytic mechanism of NK. NK not only degrades fibrin or dissolve thrombi directly, but also activates the body’s own thrombolytic system and inhibits its thrombi coagulation.

**Figure 2 foods-11-01867-f002:**
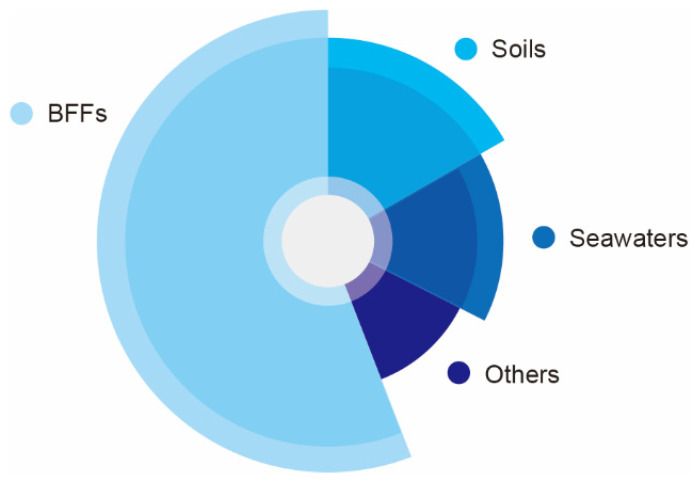
Sources of NK producing strains.

**Figure 3 foods-11-01867-f003:**
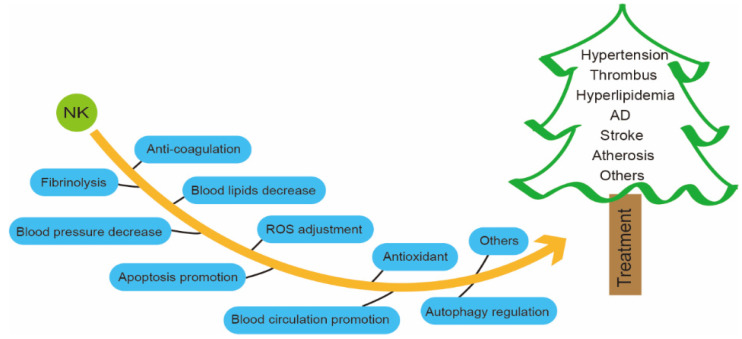
Biological functions of NK.

**Table 1 foods-11-01867-t001:** Traditional fermented soybean foods.

Geographical Location	Country	Traditional BFFs	Ref.
Asia	Japan	Natto	[24]
		Miso	[25]
	China	Douchi	[26]
		Sufu/Furu	[27]
	Korea	Chungkukjang/Chongkukjang	[18]
		Gochujang	[28]
		Doenjang	[29]
		Kanjang	[30]
	India	Kinema	[31]
		Tungrymbai	[32]
		Bekang	
		Dosa batter	[33]
	Indonesian	Moromi	[34]
		Tempeh	[35]
	Cambodia	Sieng	[36]
	Laos		
	Thailand	Thua Nao	[37]
	Nepal	Kinema	[31]
	Bhutan		
	Myanmar	Pepok	[36]
Africa	Ghana and Nigeria	Dawadawa	[38]

**Table 4 foods-11-01867-t004:** Recombinant expression of NK in different hosts.

Expression Systems	Hosts	Engineering Methods	Results	Ref.
Microorganisms	*Bacillus* spp.	Eight-proteases-gene-deficient *B. subtilis* WB800	NK was expressed at a high level of 600 mg protein per liter culture medium	[94]
		Multiple lytic genes deficient mutant *B. subtilis* LM2531	The NK activity increased about 2.6-fold compared to *B. subtilis* 168	[95]
		Eight-proteases-gene-deficient *B. licheniformis* BL10	The fermentation activity and product activity per unit of biomass of NK increased by 39% and 156% compared to *B. licheniformis* WX-02	[96]
	*E. coli*	Cytoplasmic expression NK in E. coli by cloning the *aprN* gene	The NK yield (49.3 mg/L) lower than that of *B. subtilis* YF38	[51]
		*E. coli* DH5α acted as the host for plasmid construction from	50.53% increase in NK activity compared to *B. licheniformis* DW2	[98]
		The *aprN* gene were cloned in *E. coli* pQE30	79.3 IU/mg fibrinolytic activity of NK compared to the 52.0 IU/mg from *B. subtilis* natto	[99]
	*LAB*	The recombinant plasmid pMG*thy*A-ppNK carrying the *aprN* gene in *L. bulgaricus*	Recombinant strains expressed NK intracellularly, NK activity gradually decreased to 25% after 2 h in artificial gastric juice. The highest relative activity was approximately 94% after 3 h in artificial intestinal juice	[100]
	Eucaryote	Heterologous expression cloned the *aprN* gene on vector pPICZaA-NK and transformed into *Pichia pastoris* x-33	The final fibrinolytic activity of NK is 195 U/mL	[97]
		Heterologous expression cloned the *aprN* gene in *Pichia pastoris*	Expressed high levels of NK approximately 9.5 g/L in high-density fermentation	[101]
Animals	Insects	*Rv-egfp-NK* (a recombinant baculovirus) containing the *aprN* gene expressed in *Spodoptera frugiperda* (SF-9) cells	The fibrinolytic activity of recombinant NK was 60 U/mL	[92]
Plants	Fruits	A synthetic gene (*s NK*) encoding NK was transiently expressed in *Cucumis melo* L.	The maximum fibrinolytic activity of 79.30 U/mL	[102]
	Tobacco	Synthetic genes (*sNK* and *sNKi*) encoding NK was transiently expressed in tobacco leaves	The maximum fibrinolytic activity of 16.73 U/mL	[103]

## Data Availability

Not applicable.

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
