# Peer review of "Recent Advances in Nattokinase-Enriched Fermented Soybean Foods: A Review"

_foods, 2022, doi:10.3390/foods11131867_

Round 1

Reviewer 1 Report

Title: Recent Advances in nattokinase-enriched fermented soybean foods: A Review

This review systematically discusses the recent advances in Nattokinase (NK), including fermented soybean foods, production strain sources, optimization strategies, extraction and purification procession, activity maintenance, functional evaluation, and toxicity assessment of NK. Meanwhile, it reveals the serious challenges for NK production by microorganism, as well as the prospects for NK application.

Overall, the study is well-designed and well-written, the methods are suitable and the obtained results are clearly presented and discussed.  Moreover, the conclusions have been appropriately pointed out.

Reviewer 2 Report

The present paper, regarding the Nattokinase, its presents in food products, its production by microorganisms and the activity reported in the literature result to be interesting however, can be improved.

Just some suggestions are reported below:

-        It could be interesting to dedicate a paragraph to analyze the techniques used for the detection of the NK producer strains. Are available quick methods for its detection?

-        Which is the concentration of NK in the different foods reported in the article? A specific paragraph with this information should be useful and interesting

-        Which are the concentrations needed to obtain the effects proposed in humans (for instance antithrombotic and fibrinolytic activity or anti-atherosclerotic effect, etc.)

-        Please define all the abbreviation proposed in the manuscript

-        Please check the name of the microorganisms, they have to be written in Italic (such as in lines 241, 243)
